# CCN3 (NOV) Drives Degradative Changes in Aging Articular Cartilage

**DOI:** 10.3390/ijms21207556

**Published:** 2020-10-13

**Authors:** Miho Kuwahara, Koichi Kadoya, Sei Kondo, Shanqi Fu, Yoshiko Miyake, Ayako Ogo, Mitsuaki Ono, Takayuki Furumatsu, Eiji Nakata, Takako Sasaki, Shogo Minagi, Masaharu Takigawa, Satoshi Kubota, Takako Hattori

**Affiliations:** 1Department of Biochemistry and Molecular Dentistry, Okayama University Graduate School of Medicine, Dentistry and Pharmaceutical Sciences, Okayama 700-8525, Japan; kuwahara.miho@s.okayama-u.ac.jp (M.K.); de421016@s.okayama-u.ac.jp (S.K.); fushanqi@s.okayama-u.ac.jp (S.F.); yoshiko-m@okayama-u.ac.jp (Y.M.); carinablue2004@yahoo.co.jp (A.O.); kubota1@md.okayama-u.ac.jp (S.K.); 2Department of Occlusal and Oral Functional Rehabilitation, Okayama University Graduate School of Medicine, Dentistry and Pharmaceutical Sciences, Okayama 700-8525, Japan; minagi@md.okayama-u.ac.jp; 3Department of Oral and Maxillofacial Surgery, Okayama University Graduate School of Medicine, Dentistry and Pharmaceutical Sciences, Okayama 700-8525, Japan; de422015@s.okayama-u.ac.jp; 4Department of Molecular Biology and Biochemistry, Okayama University Graduate School of Medicine, Dentistry and Pharmaceutical Sciences, Okayama 700-8525, Japan; mitsuaki@md.okayama-u.ac.jp; 5Department of Orthopedic Surgery, Okayama University Graduate School of Medicine, Dentistry and Pharmaceutical Sciences, Okayama 700-8525, Japan; matino@md.okayama-u.ac.jp (T.F.); eijinakata8522@yahoo.co.jp (E.N.); 6Department of Biochemistry, Faculty of Medicine, Oita University, Oita 879-5593, Japan; tsasaki@oita-u.ac.jp; 7Advanced Research Center for Oral and Craniofacial Sciences, Okayama University Dental School/Graduate School of Medicine, Dentistry and Pharmaceutical Sciences, Okayama 700-8525, Japan; takigawa@md.okayama-u.ac.jp

**Keywords:** cellular communication network factor 3, CCN3, NOV, primary chondrocytes, aging, oxidative stress, senescence, p21, p53, SASP

## Abstract

Aging is a major risk factor of osteoarthritis, which is characterized by the degeneration of articular cartilage. CCN3, a member of the CCN family, is expressed in cartilage and has various physiological functions during chondrocyte development, differentiation, and regeneration. Here, we examine the role of CCN3 in cartilage maintenance. During aging, the expression of *Ccn3* mRNA in mouse primary chondrocytes from knee cartilage increased and showed a positive correlation with *p21* and *p53* mRNA. Increased accumulation of CCN3 protein was confirmed. To analyze the effects of CCN3 in vitro, either primary cultured human articular chondrocytes or rat chondrosarcoma cell line (RCS) were used. Artificial senescence induced by H_2_O_2_ caused a dose-dependent increase in *Ccn3* gene and CCN3 protein expression, along with enhanced expression of *p21* and *p53* mRNA and proteins, as well as SA-β gal activity. Overexpression of CCN3 also enhanced *p21* promoter activity via *p53*. Accordingly, the addition of recombinant CCN3 protein to the culture increased the expression of *p21* and *p53* mRNAs. We have produced cartilage-specific CCN3-overexpressing transgenic mice, and found degradative changes in knee joints within two months. Inflammatory gene expression was found even in the rib chondrocytes of three-month-old transgenic mice. Similar results were observed in human knee articular chondrocytes from patients at both mRNA and protein levels. These results indicate that CCN3 is a new senescence marker of chondrocytes, and the overexpression of CCN3 in cartilage may in part promote chondrocyte senescence, leading to the degeneration of articular cartilage through the induction of p53 and p21.

## 1. Introduction

Aging is a main risk factor of senescence, which involves a decrease in biological functions such as muscle strength, nerve conduction velocity, vital capacity, and resistance to illness. In terms of cartilage, aging increases the risk of osteoarthritis (OA), which is the most common joint disease characterized by articular cartilage degeneration, and causes severe joint pain, physical disability, and impairment of quality of life. Despite recent advances in our knowledge of disease pathogenesis, treatment is still a challenge and, unlike for inflammatory joint diseases, no disease-modifying drugs are currently available for OA [1]. Although OA is not an inevitable consequence of senescence, aging is a strong risk factor for OA [2]. Chondrocytes undergo age-dependent senescence, which is characterized by a decline in the proliferative and synthetic capacity of cellular products, including their extracellular matrices; this is considered to play a significant role in the pathology of OA [3]. Although aging and OA are closely linked, they are independent processes. Several reports show the mechanism of contribution of aging to OA: age-related inflammation [4], cellular senescence (including the senescence-associated secretory phenotype (SASP)) [5,6,7], mitochondrial dysfunction and oxidative stress [8,9,10], and dysfunction in energy metabolism. This metabolic dysfunction is due to the reduced activity of 5′-AMP-activated protein kinase (AMPK) [11,12,13], which is associated with reduced autophagy. Alterations in cell signaling due to age-related changes in the extracellular matrix [14] are also of note. Improved understanding of the aging-related mechanisms that promote OA could lead to the discovery of new targets for therapies that aim to slow or stop the progression of this chronic and disabling disease.

Cellular communication network factor 3 (CCN3), previously called nephroblastoma overexpressed (NOV), is a secreted multifunctional protein involved in a variety of cellular processes, and interacts with various extracellular and transmembrane proteins [15,16,17]. Our previous studies have shown that CCN3 suppresses the proliferation and maturation of growth plate chondrocytes, regulating the expression of chondrocytic extracellular matrix (ECM) genes and Sox9 [18]. In the present study, we show for the first time that the expression of CCN3 is elevated in aging mouse and human articular cartilage. Cell cycle arrest by H_2_O_2_ demonstrated induction of *Ccn3* as well as *Col10a1* and matrix metalloproteinases mRNA, and downregulation of *Col2a1* mRNA. The overexpression of CCN3 induced promoter activity of *p21*, which has multiple *p53*-responsive elements. An in vivo mouse model of CCN3 overexpression in cartilage showed prominent degenerative changes in articular cartilage. Here, we propose that CCN3 is a novel senescence marker of chondrocytes, and overexpression of CCN3 in cartilage can be one of the factors that partly promotes chondrocyte senescence and degeneration of articular cartilage via p53 and p21.

## 2. Results

### 2.1. Increased Expression and Accumulation of CCN3 in Mouse Articular Cartilage with Aging

First, we examined the expression level of *Ccn3* mRNA in primary mouse chondrocytes from embryonic 18.5-day rib cage cartilage and postnatal two days to 37-weeks-old knee joints by quantitative RT–PCR standardized by either glyceraldehyde 3-phosphate dehydrogenase (*Gapdh*) mRNA, total RNA amount, or transmembrane protein 199 (*Tmem199*) mRNA. Since for *Gapdh,* a housekeeping gene, mRNA expression decreased with age (Appendix A), *Tmem199*, a gene reported to undergo little fluctuation in gene expression during aging [19], was also used for the normalization of *Ccn3* mRNA (Appendix A). A strong positive correlation was observed between age and *Ccn3* expression level, standardized by either *Gapdh* mRNA, total RNA amount, or *Tmem199* mRNA (Spearman’s rank correlation coefficient (*rs*): 0.77 for *Ccn3/Gapdh,* 0.77 for *Ccn3/Tmem199*, and 0.97 for *Ccn3*/µg RNA; *p* < 0.01, Figure 1A–C). Although both *Gapdh* and *Tmem199* mRNA expression showed negative correlation with age, altogether these results indicated a strong positive correlation of CCN3 expression with age in chondrocytes. For better comparison, expression of all other genes was hence standardized to *Gapdh.*

To determine whether murine chondrocytes display characteristic features of senescent cells with age, we assessed the expression of genes that arrest cell cycle, such as *p16*, *p53*, and *p21*. As shown in Figure 1D–F, the expression of these genes was positively correlated with age (*rs*: 0.77 for *p16*/Gapdh, 0.77 for *p53*/*Gapdh*, and 0.82 for *p21*/*Gapdh*; *p* < 0.01).

Expression levels of chondrocyte marker genes were also assessed: Aggrecan and type 2 collagen are the major and cartilage-specific components of the extracellular matrix. Postnatal expression of Aggrecan (*Acan*) mRNA dropped dramatically (*rs*: −0.73 for *Acan*/*Gapdh*, *p* < 0.01), while no significant age-related changes in *Col2a1* expression were observed (Figure 1G,H) in this period. The expression of *Sox9,* a transcription factor for cartilage differentiation, standardized by *Gapdh* mRNA increased with age (*rs*: 0.69 for *Sox9*/*Gapdh*, *p* < 0.01), but decreased when standardized to RNA (*rs*: −0.88 for *Sox9*/µg RNA, *p* < 0.01, Figure 1I,J; see discussion). Senescence-associated secretory phenotype (SASP) -related inflammatory factors such as Il-6 or Il-8 were elevated with age (*rs*: 0.5670076 for *Il-6/Gapdh*, and 0.5182002 for *Il-8/Gapdh*; *p* < 0.05, Figure 1K,L). Accumulation of CCN3 protein was observed by immunohistochemical staining of CCN3 in knee joint cartilage from one-, two-, and seven-month-old mice. The number of CCN3-positive chondrocytes increased in the superficial zone of seven-month-old articular cartilage (Figure 1M–O). No positive staining was observed without CCN3-specific antibody (Figure 1P).

### 2.2. Increased Expression of CCN3 in H_2_O_2_-Treated Chondrocytes

To induce cellular senescence by oxidative stress in vitro, human primary articular chondrocytes from 48 years old patient (Figure 2A) and rat chondrosarcoma-derived cells (RCS, Figure 2B) were exposed to H_2_O_2_ at various concentrations for 2 h, and the medium was changed to a normal growing medium, followed by 24 h incubation. Thereafter, total RNA was extracted and subjected to RT-qPCR analysis.

In human primary chondrocytes, *CCN3* mRNA expression increased significantly by H_2_O_2_-treatment in a dose-dependent manner (Figure 2A, * *p* < 0.05). The expression of *p21* and *p53* mRNAs was also significantly induced by H_2_O_2_ in a dose-dependent manner (Figure 2A, * *p* < 0.05), indicating cell cycle was arrested in H_2_O_2_-treated articular chondrocytes. *Col2a1* mRNA was increased at relatively lower concentration of H_2_O_2_ but increased concentration of H_2_O_2_ repressed the expression, indicating that an escaping system exists from oxidative stress. *Acan* mRNA was reduced dose dependently. Expression of *COL10A1* and *MMP13* by H_2_O_2_ was slightly induced, but it was not significant (Figure 2A).

Similarly, in RCS cells, elevated expression of *Ccn3*, *p21*, and *p53* mRNA was observed. *Col2a1* and *Acan* mRNA were also reduced in a relatively higher concentration of H_2_O_2_, however, rather lower concentration of H_2_O_2_ increased the expression in RCS cells. *Mmp13* and *Adamts5* encoding matrix metalloproteinases, which specifically degrade cartilaginous matrices, and *Col10a1*, a marker of hypertrophic cartilage, which is induced during osteophyte formation, were significantly induced by treatment with H_2_O_2_ in RCS cells (Figure 2B, * *p* < 0.05), however, at highest concentration of H_2_O_2_, *Mmp13* mRNA was reduced, indicating that the maximum concentration of H_2_O_2_ to induce *Mmp13* was lower than that for *Col10A1* or *Adamts5* mRNA in RCS cells.

CCN3 and p53 production in H_2_O_2_-treated RCS cells was also monitored by Western blot using an anti-CCN3 or anti-p53 antibody. CCN3 was accumulated in H_2_O_2_-treated RCS after 24 h incubation in a dose-dependent manner, but 48 h after the H_2_O_2_ treatment at the highest concentration, the amount of CCN3 as well as beta-actin decreased (Figure 2C). In contrast, p53 was accumulated in H_2_O_2_-treated RCS after 48 h incubation in a dose-dependent manner (Figure 2D), but not after 24 h incubation (data not shown), indicating p53 protein was accumulated after CCN3 protein was produced. Specificity of antisera against CCN3 was monitored by expressing green fluorescent protein (GFP)-fused CCN3 protein with or without Flag epitope using expression vector and those molecular weights were compared with Western blot signals obtained using anti GFP antibody (Figure 3A,B).

Cellular senescence induced by H_2_O_2_ treatment in RCS cells was also estimated by monitoring senescence associated (SA) β-gal activity. Thirty-six hours treatment with 100 and 200 µM H_2_O_2_ exhibited SA-β-gal positive senescent cells in RCS cells (Figure 2E).

### 2.3. Overexpression of CCN3 Stimulates p21 Promoter in RCS Cells

To understand whether elevated CCN3 expression is a result of aging, or whether elevated CCN3 expression induces aging, we monitored the activity of a *p21* promoter with a plasmid construct harboring p53 binding sites after overexpression of CCN3 in RCS cells. CCN3 overexpression in RCS cells was confirmed by Western blot with both anti GFP antibody and CCN3 antibody (Figure 3A,B, ▶: GFP-CCN3 (67 kDa), ▷: GFP-2xFlag-CCN3 (69 kDa)). Overexpression of GFP-tagged both CCN3 and Flag-CCN3 was also monitored with green fluorescent protein (data not shown). The *p21* promoter activity was monitored two days after transfection. Increased *p21* promoter activity was observed by cotransfection of the pEGFP-CCN3 vector in a dose-dependent manner (Figure 3C, *: *p* < 0.05). These results suggested that overexpression of CCN3 in RCS cells induces p21 expression through induction of p53.

### 2.4. Treatment with CCN3 Stimulates p21, p53 mRNA Expression in Mouse Primary Chondrocytes and RCS Cells

To see whether exogenously added CCN3 has ability to induce cell cycle arrest or not, recombinant (r) CCN3 was added to mouse primary articular chondrocytes from 2 W old knee joints and RCS cells. After 24 h incubation, total RNA was harvested and analyzed.

As a result, *p21* was used to standardize total amounts of cDNA; in some experiments, transmembrane protein 199 and *p53* mRNA expression levels were significantly increased in both mouse primary chondrocytes (Figure 4A) and RCS cells (Figure 4B) by the addition of rCCN3 (* *p* < 0.05, ** *p* < 0.01).

### 2.5. Cartilage-Specific CCN3 Overexpression Induced Degradative Changes in Articular Knee Joints and Rib Chondrocytes In Vitro

To further verify the function of CCN3 in vivo, we generated CCN3-overexpressing mice under the control of a 6 kb-*Col2a1* promoter that included a cartilage-specific enhancer in the first intron of the *Col2a1* gene. The CCN3 gene was fused with the gene of green fluorescent protein (GFP), and IRES-LacZ was also built in to trace the transgene expression. The activity of beta-galactosidase in cartilage could be traced in up to 18-month-old knee joints (Figure 5A,B). Interestingly, Safranin-O staining of two-month-old knee joints showed degenerative changes, such as thinning of articular cartilage layers and surface roughening in transgenic mice (Figure 5C). Degenerative changes in articular cartilage were more severe in seven-month-old mice (Figure 5D). In addition, immunohistochemical staining for the aggrecan neoepitope in two-month-old knee joints showed positive staining in CCN3-overexpressing articular cartilage only, indicating that overexpressed CCN3 in cartilage induced degenerative changes in knee joints (Figure 5E).

Total RNA was obtained from the cultured primary chondrocytes of embryonic 18.5-day-old ribcages and those from the ribcage and articular cartilage of three-month-old transgenic mice. Even in 18.5-day-old *Ccn3* transgenic rib chondrocytes, increased expression of *p53*, *p21*, *Adamts5*, and *Mmp13* mRNA, and decreased expression of type II collagen was observed. The *Acan* expression level increased unexpectedly (Figure 5F, *p* < 0.01). In three-month-old *Ccn3* transgenic articular and ribcage cartilage, *p21*, inflammatory factors of *Il-6*, *Il-8*, and *Tnfα* were induced; in contrast, *Col2a1*, *Aggrecan*, and *Sox9* mRNA were downregulated (Figure 5G, *p* < 0.01). However, *p53* mRNA did not show a significant difference (data not shown).

### 2.6. Elevation of CCN3, p53, and p21 mRNA Expression in Human Articular Cartilage with Age

We also analyzed the expression level of CCN3 in human cartilage samples obtained after joint surgery. Cartilage samples were obtained from 14 patients aged 19 to 91 years who were relatively healthy in those years, without osteoarthritic degenerative changes. RT-qPCR analysis of primary cultured chondrocytes from articular cartilage showed a strong positive correlation between age and *CCN3*, *p53*, and *p21* expression levels (*rs*: 0.8030808 for CCN3/GAPDH, 0.7018706 for p53/GAPDH, 0.8294835 for p21/GAPDH, *p* < 0.01, Figure 6A), which is similar to the results in mice (Figure 1). As note, similar to mouse chondrocytes, human articular chondrocytes also showed dramatically decreased *GAPDH* expression (see Appendix A). *CCN3* mRNA level was also normalized with amount of RNA, and *CCN3* mRNA still showed strong positive correlation between age (data not shown). Western blot analysis of cell lysates from 19- and 72-year-old primary articular chondrocytes showed CCN3 bands with different molecular weights (Figure 6B, see also Section 3). Immunohistochemical staining of CCN3 revealed strong positive staining in the superficial layer of femur articular cartilage in 74- and 81-year-old patients (Figure 6C).

## 3. Discussion

Osteoarthritic degenerative changes in articular cartilage are highly associated with chondrocyte senescence caused by aging. In this study, we explored the role of CCN3 in aging in articular cartilage. We observed significant induction of CCN3 in aging chondrocytes and in cells after oxidative stress- induced senescence. Mouse chondrocytes from older adults exhibited many changes in gene expression that are typical of senescent cells, such as induction of p21 and p53, mediators of cell cycle arrest, Il-6 and Il-8, as SASP factors, and reduction of the extracellular matrix gene, *Acan*. Our data support the hypothesis that the physiological stresses of aging induce the expression of cell cycle arrest genes, even in a hyporeplicative cell type such as chondrocytes.

A quantitative assessment of gene expression in aging cells is hampered by the observation that aging is associated with changes in gene expression levels that affect cellular functions, including cellular metabolism; therefore, housekeeping genes are also not always stably expressed [19]. Recently, transmembrane protein 199 (TMEM199) has been described as a new candidate reference gene to normalize the RT-qPCR data of senescent cells [20]. In our study, a significant increase in the expression of *Ccn3* along with age was found by normalization, not only with *Gapdh* and RNA amount, but also with *Tmem199*. Increased expression of CCN3 with age was also observed in human articular cartilage. Changes in the expression of many marker genes in chondrocytes with age have been reported with or without cartilage degeneration [21,22]. The reported changes, however, have to be taken with care since many of the genes used as internal control show decreased expression along the age. For example, in this report, *Col2a1* did not show expression changes with age when standardized to *Gapdh*. *Sox9* mRNA expression showed positive correlation with age when standardized to *Gapdh*, in contrast, negative correlation was observed when normalized to total RNA.

Western blot analysis of human articular chondrocyte cell lysates showed CCN3 bands with different molecular weights between young and aged chondrocytes; CCN3 in aging chondrocytes showed a higher molecular weight, suggesting that CCN3 may undergo some modifications with age. Whether this modification changed the CCN3 activity or stability needs to be further elucidated.

Cellular senescence is a complex phenotype characterized by two parameters: durable cell cycle arrest induced by cellular stress; and the production of a set of pro-inflammatory molecules and matrix metalloproteinases known as the senescence-associated secretory phenotype (SASP) [23,24,25]. Artificially induced cellular senescence with oxidative stress by H_2_O_2_ in both human articular chondrocytes and RCS cells induced *CCN3* mRNA in a dose dependent manner. Genes of cell cycle arrest, *p21* and *p53* mRNA were also induced in both cells dose dependently. Induction of CCN3 and p53 proteins by H_2_O_2_ was also examined in RCS cells. Activation of SA-β gal was also detected in H_2_O_2_ treated RCS cells demonstrating H_2_O_2_ induced cellular senescence in cultured chondrocytes inducing CCN3 expression. Changing of mRNA expression of extracellular matrix was observed, but not in a dose-dependent manner. Expression of *COL2A1* and *Acan* mRNA rather enhances in lower concentration of H_2_O_2_ in both primary chondrocytes and RCS cells, may indicating that mild damage of chondrocytes by oxidative stress rather induces synthesis of cartilaginous matrices observed in early stage of osteoarthritis [26]. *Col10a1*, *Mmp13*, and *Adamts5* mRNA found in osteoarthritic cartilage [27,28] were also strongly induced in H_2_O_2_ treated RCS cells, while a weaker response was seen in primary chondrocytes. This may be the reason that immortalized cell line could not survive when cell cycle was arrested, however, primary cultured cells go through maturation before death.

Not only elevated expression of CCN3 during aging and in induced senescent cells, but also overexpression of CCN3 or addition of recombinant CCN3 protein to cultured chondrocytes in vitro was found to induce cell cycle arrest by upregulating *p53* and *p21*.

Furthermore, CCN3 overexpression in articular cartilage in vivo not only affected cell cycle regulation but also induced characteristic features of senescence. In this report, severe degenerative changes were observed in CCN3-overexpressing articular cartilage. For the gene expression analysis, we isolated primary chondrocytes from rib cages of embryonic 18.5 days, and rib cage cartilage of 3 month old transgenic and wild type mice of the same littermates. Even in 3 month rib cage cartilage, CCN3 induced *Il-6*, *Il-8*, and *Tnfα* which are considered as general SASP factors and also osteoarthritic proinflammatory degenerative markers in articular cartilage. In addition, decreased expression of *Col2a1* and *Acan* mRNA was observed, which is consistent with a previous result [18]. However, *Mmp13* and *Adamts5* mRNA were not upregulated in three-month old rib chondrocytes in contrast to articular cartilage. p53 mRNA was not also elevated in three-month old rib RNA sample, this may be due to the stably translated p53 protein level or p21 protein level. In E18.5-*Ccn3* transgenic chondrocytes, elevated expression of *p53*, *p21*, *Adamts5*, and *Mmp13* mRNA was observed, but SASP factors such as *Il-6, Il-8*, and *Tnfα* were not induced (data not shown), indicating that the induction of those factors only occurs after birth.

A physiological interaction between CCN3 and p53 has not been reported so far. It is possible that p53 regulates *Ccn3* transcription or that CCN3 protein may upregulate another transcription factor(s) which is induced along aging to upregulate p53. It is also possible that CCN3 protein may itself form complexes that regulate cell cycle. Further studies will be necessary to clarify this problem.

Two different lines of CCN3-deficient mice have been reported: CCN3-deficient mice in which exons 1 and 2 and part of exon 3 are replaced by the neomycin cassette showed enhanced neointimal hyperplasia in response to injury. This is caused by repressing the inhibitory effects of CCN3 on vascular smooth muscle cell proliferation, which is mediated by induction of p21 and p15 [29]. This report is in line with our observation that CCN3 induced p21 and p53 expression in chondrocytes. In our data, there was an increase in the expression level of p21 but not of p53 in three-month-old CCN3-overexpressing cartilage. However, our finding that both treatment with recombinant CCN3 and CCN3 overexpression significantly increased the p53 expression indicates that p53 regulation by CCN3 also occurs in chondrocytes. The other CCN3-deficient mice caused by deletion of exon 3 showed skeletal abnormalities in some joints and heart defects [30]; 12-month-old male knee joints showed some degenerative changes, but female joints did not [31]. Since these CCN3-deficient mice were able to express the first 1 and 2 domains coded by exon 1 and 2, the phenotypes of these mice must be interpreted with care. Whether those mice show prolonged life or anti-aging phenotype, or not, needs to be analyzed.

Although the precise molecular mechanism of regulation of the cell cycle by CCN3 is still unclear, there are several reports that CCN3 regulates NOTCH1 signaling [32]. The detailed signaling mechanism of cell cycle control by CCN3 in adult articular cartilage should be investigated in future studies.

The accumulation of senescent cells with aging contributes to age-related tissue dysfunction [33]. However, cellular senescence has also been considered as a protective system against tumorigenesis. CCN3 has antiproliferative activities in several tumor cell lines [34,35,36]. In addition, CCN3 was previously shown to be induced by glucose starvation [37]. Another report suggests that recombinant CCN3 ameliorates 1L-1β induced metalloproteinases in cultured chondrocytes by the activation of PI3K/AKT/mTOR pathway [38,39]. Taking these facts into consideration, it is assumed that the observed CCN3 induction may be a protective response against multiple stimuli such as age-related oxidative stress and nutrition shortages.

## 4. Conclusions

CCN3 can be a new marker of chondrocyte senescence, and CCN3 accelerates cellular senescence via induction of *p53* and *p21*.

## 5. Materials and Methods

### 5.1. Cell Cultures

Mouse primary articular chondrocytes were isolated from neonatal day 2 to 37-week-old articular cartilage; primary ribcage chondrocytes were isolated from embryonic 18.5-day- and three-month-old C57BL/6J mice, as previously described [40,41].

Human articular chondrocytes were obtained from patients with informed consent upon orthopedic surgery in a traumatic clinical case approved by the Ethics Committee of Okayama University Graduate School of Medicine, Dentistry, and Pharmaceutical Sciences (#K1506-018). Pieces of knee articular cartilage were treated similarly to mouse articular cartilage, except for overnight digestion with 2% collagenase. Rat chondrosarcoma-derived cells (RCS) were provided by Dr. Benoit de Crombrugghe (The University of Texas MD Anderson Cancer Center, Houston, TX, USA) [42]. Cells were cultured in Dulbecco’s modified Eagle’s medium (DMEM) containing 10% fetal bovine serum (FBS) at 37 °C with 5% CO_2_. Before the addition of recombinant mouse CCN3 (see Section 2.6), RCS cells were cultured with DMEM containing 0.5% FBS for 24 h. All animal experiments were conducted in accordance with the institutional rules under the approval from the Animal Care and Use Committee of Okayama University, Okayama, Japan (OKU-2018903, OKU-2018023, OKU-2020328).

### 5.2. Isolation of Total RNA

Total RNA from cultured cells was extracted and purified using a RNeasy Mini Kit (Qiagen, Hilden, Germany). For cartilage, tissues were homogenized with ISOGEN (Nippon Gene, Toyama, Japan), and total RNA was further purified by a RNeasy Mini Kit, according to the manufacturer’s instructions.

### 5.3. Real-Time Polymerase Chain Reaction

Total RNAs were reverse-transcribed by a PrimeScript RT-reagent Kit (Takara Bio, Shiga, Japan). Subsequent quantitative real-time PCR was performed using SYBR^®^ Green Realtime PCR Master Mix (Toyobo Osaka, Japan) with a StepOnePlus™ (Applied Biosystems, Basel, Switzerland) Real-Time PCR System. The primers used are listed in Table 1, Table 2 and Table 3. Expression of *Gapdh* was used to standardize total amounts of cDNA; in some experiments, transmembrane protein 199 (*Tmem199*) or the amount of total RNA was used for standardization.

### 5.4. Histology

Knee joints were fixed with 10% formaldehyde/PBS, demineralized with 0.5 M EDTA, dehydrated, and embedded in paraffin. After the tissues were embedded in paraffin, serial frontal sections of 5 µm thickness were cut through the knee joints.

### 5.5. Immunohistochemistry

Immunohistochemistry was done as reported previously [40,41]. Sections of knee joints were treated with bovine testicular hyaluronidase (10 mg/mL) for 30 min at room temperature for epitope retrieval, followed by 10% H₂O₂ in methanol for 10 min to inactivate endogenous peroxidase, and then were immunostained with anti-CCN3 (provided by Dr. Takako Sasaki), or anti-aggrecan neo-epitope (Novus Biologicals, Littleton, CO, USA) antibody. The signal was enhanced with Envision+ Dual Link System Peroxidase (Dako, Glostrup, Denmark) and developed with DAB (2 mg/mL). In some experiments, the sections were counter-stained with methyl green.

### 5.6. Recombinant Mouse CCN3 and CCN3 Antibody

Protein coding region of mouse *Ccn3* cDNA (NM_010930.5) including signal peptide was cloned into the pCEP vector. The expression construct was transfected into 293 EBNA cells and serum-free conditioned media were collected. The conditioned media were loaded onto DEAE cellulose (DE52) equilibrated with 0.05 M Tris-HCl, pH 8.6 and eluted with a linear 0–0.5 M NaCl gradient in DEAE buffer. Fractions containing CCN3 protein were concentrated by ammonium sulfate, and further purified on Superose 12 HR 15/60 column eqilibrated with 0.2 M ammonium acetate. The CCN3 protein was dialyzed against PBS before addition to the cells.

CCN3 antiserum was obtained by immunizing rabbits with mrCCN3. For Western blot, 1:500 dilution, and for immunohistochemistry, 1:50 dilution were used. Specificity of antiserum was monitored by comparison of the staining signal between CCN3 tg and wild type cartilage (Appendix A).

### 5.7. Treatment with H₂O₂

After RCS cells reached confluence, the cells were exposed to 100–400 μM H₂O₂ for 2 h. The cells were washed with DMEM and replenished with a fresh medium containing 10% FBS. Total RNAs were collected after 24 h incubation, and cell lysates were collected after 24 and 48 h incubation.

### 5.8. Western Blots

Cultured cells were washed with PBS and harvested with 10 mM Tris–HCl (pH 8.0) in 150 mM NaCl, 1% Triton X-100, and 0.1 mM phenylmethylsulphonyl fluoride. After the removal of precipitates, 5 µg of total protein were subjected to Western blot analysis. Primary antibodies against CCN3 (provided by Dr. Takako Sasaki, Oita University) and β-actin (Sigma, St Louis, MO, USA) were used.

### 5.9. Plasmids

For overexpression of CCN3 in RCS cells, mouse *Ccn3* cDNA was ligated into a pEGFP‒N1 vector (Clontech). For *p21* reporter gene assay, a firefly luciferase reporter vector, pGL3‒PG12S (kindly provided by Dr. Inoue, Wakayama Medical University) containing 12× multiple p53-binding sites upstream of the *SV40* promoter, and the pGL3‒Renilla vector, harboring renilla luciferase, were used.

### 5.10. p21 Promoter Reporter Gene Assay

RCS cells were seeded in 12-well plates at a density of 8.0 × 10⁴ cells/well and incubated for 24 h. The medium was changed, and pGL3‒PG12S, pGL3‒Renilla, and various amounts of pEGFP‒CCN3 were cotransfected into the RCS cells with PEI-MAX (kindly provided by Dr. Takarada, Okayama University) according to the manufacturer’s instructions. Twenty-four hours after DNA transfection, luciferase activities were measured using a Dual-Glo Luciferase Assay System (Promega). The transfection efficiency was normalized to Renilla luciferase activity expressed from pGL3‒Renilla. Mean values were calculated from the results of three independent experiments.

### 5.11. Generation of Transgenic Mice

*Ccn3* transgenic (*Ccn3*^Tg^) mice were generated and maintained in the C57BL/6J background, in which murine Ccn3 fused with green fluorescent protein (GFP, Invitrogen, Carlsbad, CA, USA) and IRES (internal ribosome entry site)‒LacZ was driven under the control of the 6-kb *Col2a1* promoter‒enhancer. The sequences of PCR primers used for genotyping were 5′-GCATCGAGCTGGGTAATAAGCGTTGGCAAT-3′ and 5′-GACACCAGACCAACTGGTAATGGTAGCGAC-3′, which detects a fragment of *LacZ*.

The knee joints and rib cartilage of sacrificed mice were isolated for RT-qPCR and histology. All mice were housed in filter-top cages with paper-chip bedding under standard pathogen-free conditions. The mice were fed a standard diet with tap water provided ad libitum.

### 5.12. X-Gal Staining

To trace transgene expression in articular cartilage, beta-galactosidase activity was monitored from 1-, 2.5-, 7-, and 18-month-old transgenic knee joints using X-gal (5-bromo-4-chloro-3-indolyl-D-galactopyranoside, Roche, Basel, Switzerland) as a substrate, as described before [41]. For staining of knee joints, skin and muscle were removed before fixation. X-gal-stained knee joints were postfixed overnight in 4% formaldehyde, dehydrated, and embedded in paraffin. Seven-micrometer-thick sections were prepared by standard methods and counterstained with eosin.

For monitoring SA-β gal activity, the cells were fixed with 4% formaldehyde, and after washing with PBS, the cells were stained with citric acid/sodium phosphate buffer (pH 6.4) containing 5 mM potassium ferricyanate, 5 mM ferrocyanate, 150 mM NaCl, 2 mM MgCl2, 1 mg/mL X-gal for 2 h [43].

### 5.13. Safranin-O Staining

Safranin-O-fast green staining were used to analyze proteoglycan in articular cartilage.

### 5.14. Statistic Analysis

Data are presented as means ± standard deviation. The Spearman correlation coefficient was used to explore possible associations between variables. Tukey’s multiple comparison tests was used to determine the overall differences between groups. Unpaired Student’s *t*-tests were used for comparisons between the control and subjects to determine significant differences. All calculations were performed using statistical web software (http://www.gen-info.osaka-u.ac.jp/testdocs/tomocom/, Osaka University, Japan). Values of *p* < 0.05 were considered statistically significant.

## Figures and Tables

**Figure 1 ijms-21-07556-f001:**
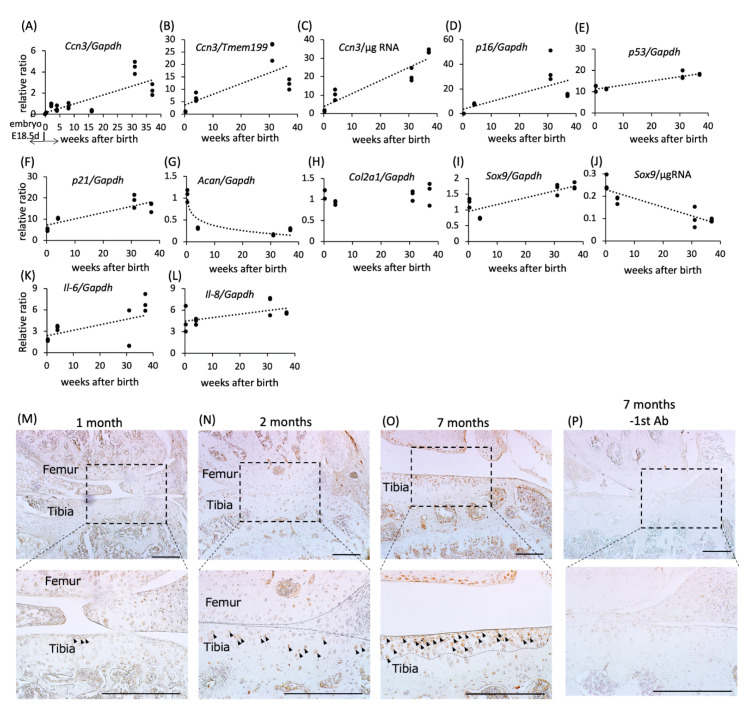
Comparison of mRNA expression levels of (**A**–**C**) *Ccn3*, (**D**) *p16*, (**E**) *p53*, (**F**) *p21*, (**G**) *Acan*, (**H**) *Col2a1*, (**I**,**J**) *Sox9*, (**K**) *Il-6*, and (**L**) *Il-8* in articular chondrocytes of different ages (embryo E18.5 represents RNA from ribcage chondrocytes of embryonic day 18.5). The expression level of *Ccn3* mRNA was standardized with (**A**) *Gapdh*, (**B**) *Tmem199*, a gene with little fluctuation in expression during aging, and (**C**) RNA amount. The expression level of *Acan* mRNA dramatically decreased with age, but *Col2a1* mRNA expression did not. *Sox9* mRNA expression increased when it was standardized by *Gapdh*; however, the expression level/μg RNA decreased with age. (**K**,**L**) *Il-6* and *Il-8*, as senescence-associated secretory phenotype (SASP) factors, also showed increased expression with age (*p* < 0.05). Either rib cage (E.18.5) or knee articular (postnatal) cartilage was pooled from 8 embryos or 4 postnatal animals and seeded into at least 3 dishes. RNA from each dish was analyzed with triplicate by real time PCR, and the average of the value from one RNA sample was plotted as one ‘dot’. (**M**–**P**) Immunohistochemical staining of CCN3 in one-, two-, and seven-month old mouse articular cartilage. CCN3-positive chondrocytes (arrow) increased in the superficial layer of articular cartilage (dotted circle) of seven-month-old compared with one-month-old mice. No signal was observed without CCN3-specific antibody (Figure 1P, -1st Ab). Scale bar: 200 µm.

**Figure 2 ijms-21-07556-f002:**
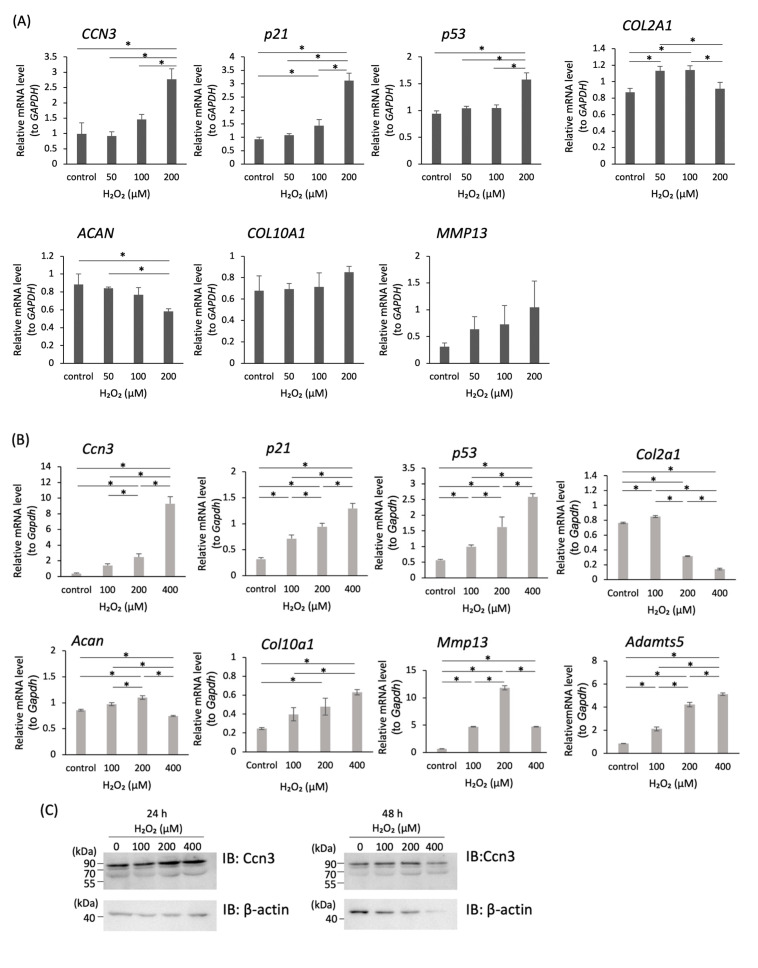
Increased expression of *Ccn3* mRNA and protein in H_2_O_2_-treated chondrocytes. (**A**) Human primary articular chondrocytes isolated from 48 years old patient and (**B**) RCS cells were treated with increased concentration of H_2_O_2_ as indicated for 2 h to induce artificial senescence by oxidative stress, followed by changing to the normal growing media without H_2_O_2_. Total RNA was collected 24 h after changing media, and (**A**) *CCN3, p21, p53, COL2A1, ACAN, COL10A1, MMP13* mRNA, and (**B**) *Ccn3*, *p21*, *p53*, *Col2a1*, *Acan*, *Col10a1*, *Mmp13*, and *Adamts5* mRNA levels were monitored and standardized with *Gapdh*. The data represent the mean ± SD (*n* = 3 individual cultures); * *p* ≤ 0.05. (**C**) Cell lysates were collected both 24 and 48 h after changing media and CCN3 protein level was monitored with an anti-CCN3 antibody. (**D**, left) Cell lysates were collected 48 h after changing media and p53 protein level was monitored with an anti-p53 antibody. (**D**, right) Band intensity of p53 was standardized to βactin. (**E**) Colorimetric detection of senescence associated β galactosidase in H_2_O_2_-treated RCS cells. RCS cells were treated with H_2_O_2_ at indicated concentration for 2 h and after media change further incubated for 36 h. bar: 200 μm.

**Figure 3 ijms-21-07556-f003:**
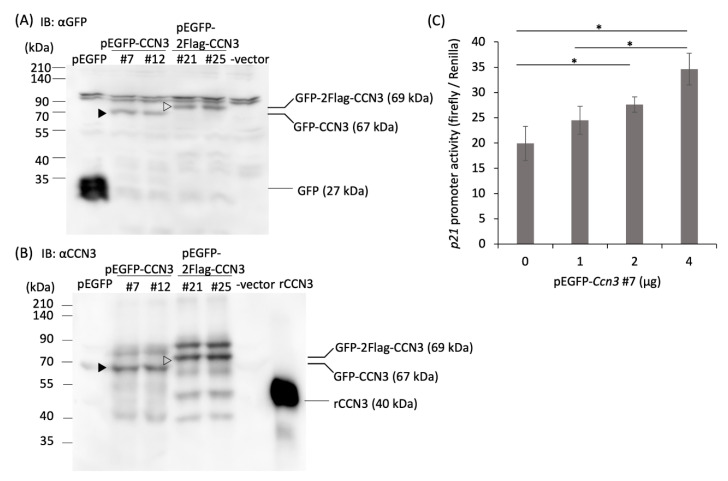
Induction of *p21* promoter activity by overexpression of CCN3 in RCS cells. Detection of GFP-CCN3 (▶) and GFP-2Flag-CCN3 (▷) overexpression with (**A**) an anti-GFP antibody, and (**B**) anti-CCN3 antibody. Two days after transfection of pEGFP-CCN3 and pEGFP-2Flag-CCN3 vector (2 clones each) by electroporation, proteins were extracted from the cells and subjected to Western blot analysis. For the p21 promoter assay in (**C**), we used pEGFP-CCN3. (**C**) pEGFP-CCN3 vector, pGL3-PG12S containing *p21* promoter with 12× multiple p53-binding sites, and pGL3-Renilla vector were transfected by electroporation, and two days after transfection, firefly luciferase activity was monitored and standardized with Renilla luciferase activity (The data represent the mean ± SD (*n* = 3 individual cultures); * *p* < 0.05).

**Figure 4 ijms-21-07556-f004:**
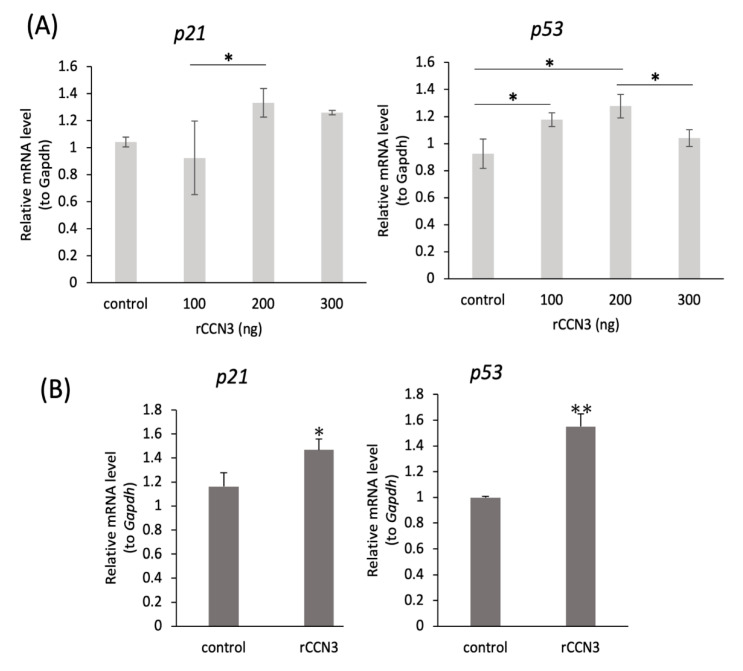
Stimulation of *p21* and *p53* mRNA expression by recombinant CCN3 treatment in (**A**) mouse primary articular chondrocytes from 2w knee joints and (**B**) RCS cells. Recombinant CCN3 was added to the media at indicated concentrations (**A**) and 100 µg/mL (**B**). Total RNA was harvested after 24 h incubation. The data represent the mean ± SD (*n* = 5 individual cultures). * *p* < 0.05, ** *p* < 0.01.

**Figure 5 ijms-21-07556-f005:**
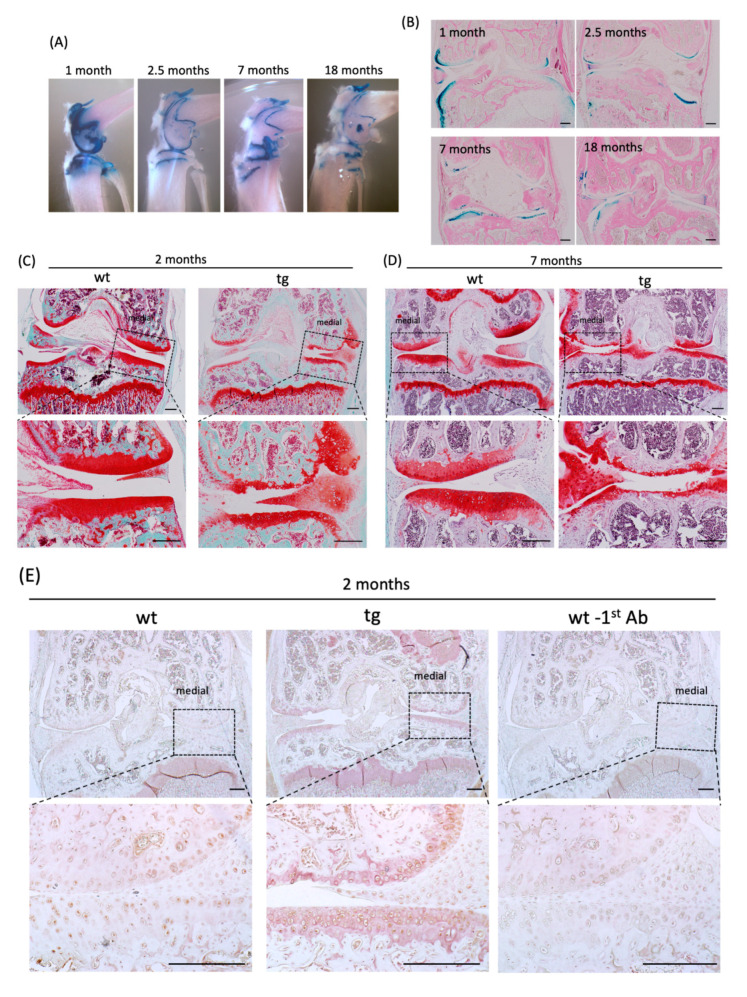
Degradative changes in knee joints and rib chondrocytes as a result of cartilage specific CCN3 overexpression in mice. (**A**) Whole mount LacZ staining of transgenic knee joints harboring a *Col2a1* promoter-GFP-CCN3-LacZ construct (medial view). Beta-galactosidase activity was traceable up to 18 months after birth. (**B**) Histological analysis of (**A**) and counterstained with eosin. (**C**,**D**) Safranin-O staining of two-month-old (**C**) and seven-month-old (**D**) *Ccn3*^tg^ mice. (**E**) Immunohistochemical staining of Aggrecan Neoepitope in two-month-old *Ccn3*^tg^ mice knee articular cartilage. (wt -1st Ab) Negative control without 1st antibody of wt sections. (**F**,**G**) Analysis of gene expression of extracellular matrices and matrix metalloproteinases in chondrocyte RNA obtained from (**F**) rib cage chondrocytes of embryonic 18.5-day-old and (**G**) ribcages of three-month-old of tg mice. (**F**) Increased expression of p53, p21 and decreased expression *Col2a1* mRNAs, were observed. (**G**) High expression levels of p21 and decreased expression levels of *Col2a1*, *Acan*, and *Sox9* were observed. Furthermore, the expression levels of inflammatory factors *Il6*, *Il8*, and *Tnfα* were increased. The *p53* expression did not show significant differences (data not shown). Scale bar: 200 µm.

**Figure 6 ijms-21-07556-f006:**
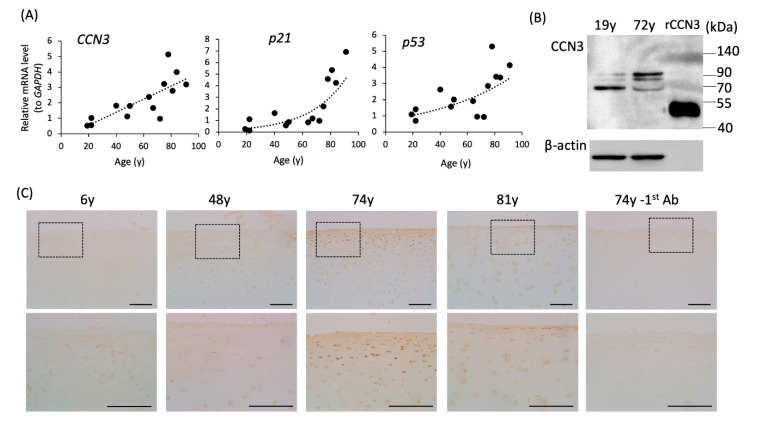
Elevated expression of *Ccn3*, *p53*, and *p21* mRNA in aged human articular cartilage. (**A**) Comparison of mRNA expression levels of *CCN3*, *p53*, and *p21* in the primary culture of human articular chondrocytes at different ages. The expression level of *Ccn3*, *p53*, and *p21* mRNA was standardized to *Gapdh*. From one patient, articular cartilage was collected and digested, and cultured in multiple dishes. Total RNA was collected at least 3 dishes individually, and real time-PCR was performed in triplicates for each RNA sample; one ‘dot’ indicates average of gene expression level from one patient. (positive correlation, all *p* < 0.01). (**B**) Western blot analysis of CCN3 protein in primary culture of the articular chondrocytes (8 μg protein/lane) from different ages (y) as indicated. (**C**) Immunohistochemical staining of CCN3 in human articular cartilage (tibiae) of several ages. Top: lower magnification; bottom: higher magnification of the black square in the top. Scale bar: 200 µm.

**Table 1 ijms-21-07556-t001:** Primer sequences for mouse tissues.

Gene	Forward	Reverse
*Ccn3*	cagaccccaacaaccagact	acttctctccgttgcggtaa
*p16*	gaactctttcggtcgtaccc	agttcgaatctgcaccgtagt
*p21*	gaacatctcagggccgaaaa	tgcgcttggagtgatagaaatc
*p53*	tcttatccgggtggaaggaaa	ggcgaaaagtctgcctgtctt
*Aggrecan*	gaggagagaactggagaag	ggcgatagtggaatacaa
*Col2a1*	tggtggagcagcaagagcaa	cagtggacagtagacggaggaaa
*Col10a1*	tgctgcctcaaataccctttct	tggcgtatgggatgaagtattg
*Sox9*	atctgaagaaggagagcgag	tcagaagtctccagagcttg
*Il-6*	ggagcccaccaagaacgata	tcccaagaaggcaactggat
*Il-8*	aatttccaccggcaatgaag	cccgaattggaaagggaaat
*Mmp13*	tcctcggagactggtaatgg	tgatgaaacctggacaagca
*Adamts-5*	ggcatcattcatgtgacac	gcatcgtaggtctgtcctg
*Tnfα*	aggggattatggctcagggt	tgagtccttgatggtggtgc
*Tmem199*	aaatggcgtcttccttgcttgc	atcactgcccgcgtgtttctt
*Gapdh **	caatgaccccttcattgacc	gacaagcttcccgttctcag

* Common to mice, RCS, and humans.

**Table 2 ijms-21-07556-t002:** Primer sequences for RCS.

Gene	Forward	Reverse
*Ccn3*	tgaagtctctgactccagcatt	tggctttcagggatttcttg
*p21*	caaagtatgccgtcgtctgttc	gaagtcaaagttccaccgttctc
*p53*	tccgactataccactatccactaca	ggcacaaacacgaacctcaaa
*Aggrecan*	agaatcaagtggagccgtgtt	ggggatggctggatagttgg
*Col2a1*	ctcacgccttcccattgttg	gttgttttggggttgagggttt
*Col10a1*	acctcccaccccattcca	acccactattgctgctcactc
*Mmp13*	tgcggttcactttgaggaca	tcttcttctatgaggcgggga
*Adamts-5*	tagaccctacagcaactccgt	cctccacacactccacactt

**Table 3 ijms-21-07556-t003:** Primer sequences for humans.

Gene	Forward	Reverse
*CCN3*	ggagcgcgctataaaacctg	tcccctctcgcttttaccaa
*p21*	tggggctgggagtagttgt	gctggaaggtgtttggggt
*p53*	ggtcggtgggttggtagttt	gtgtgggatggggtgagattt

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
