# Peer review of "CCN3 (NOV) Drives Degradative Changes in Aging Articular Cartilage"

_ijms, 2020, doi:10.3390/ijms21207556_

Round 1

Reviewer 1 Report

The paper by Kuwahara et al. entitled “CCN3 (NOV) Drives Degradative Changes in Aging Articular Cartilage” presents an interesting study on the role of the CCN3 (NOV) protein in certain degenerative changes produced in articular cartilage with age. The study is well designed and analyses CCN3 expression as well as its effect on numerous senescence and inflammatory markers in a CCN3 overexpressing transgenic mice model. The results obtained are interesting and support the hypothesis proposed by the authors, however there are some aspects of content and form that must be improve and clarified before the paper is suitable for publication.

The following aspects or points should be improved or clarified:

  • The authors analyse an extensive number of molecular markers related to different aspects of cartilage biology, some of these are specifically mentioned and their function in joint homeostasis is also mentioned, however, of others, only their abbreviation appears without being mention nothing or little about them, therefore it would be interesting that the authors include in the paper a table with all the markers analysed, indicating their molecular characteristics and their function in articular cartilage homeostasis.
  • In section 2.1 of results, lines 90 to 94, the authors mention that due to the decrease that the GADPH gene suffers with age, they propose the use of a new housekeeping gene, Tmem199, to normalize the expression of Ccn3, observing differences in the relative ratio between normalization with GADPH and Tmem199, however, the expression of the rest of the studied molecular markers that is presented in figure 1, is normalized with respect to GADPH expression, this fact creates some confusion, at least for me, for this reason, I think that it should be better clarified by the authors in the text.
  • In section 2.2 of results, the authors show the variations in the expression of Ccn3 and other markers in response to the exposure of rat chondrosarcoma derived cells (RSC) to different concentrations of hydrogen peroxide in order to mimic oxidative stress conditions. The results obtained show variations that could be expected, except for some results, as the increase of Acan up to 200 µM of H2O2, and the decrease of Mmp13 with 400 µM of H2O2. How do the authors explain these results? Are the H2O2 concentrations used, comparable with the in vivo cellular conditions of an oxidative stress situation? Can the results obtained in these experiments be extrapolated to the in vivo situation in senescent articular cartilage chondrocytes, taking into account that they have been obtained in an established chondrosarcoma cell line? Why have these experiments not been done on primary chondrocyte cultures from young adult animals? The authors should address all these aspects, which in my opinion are of great interest, in the discussion of the paper. In relation to this last aspect, it would be interesting to propose this test in primary cultures of articular cartilage chondrocytes extracted from young adult animals and compare the results with the native expression in cultures, or directly in articular cartilage samples obtained from old animals to validate the test.
  • Figures 3A and 3B are not well explained in the corresponding figure caption, at least it is difficult for me to interpret the results shown, and the authors should try to present it more clearly to the reader.
  • In Figure 5, in the results section, the authors present the changes observed in the transgenic mice model that overexpressed CCN3. In figure 5B, the regions of overexpression of CCN3 at the joint level are shown at the histological level using the LacZ gene present in the transgene. The images shown are counterstained with eosin (E) as indicate in the section 4.4 of M&M, but it is not indicated in the figure caption. Figures 5C and 5D show the morphological changes at the knee joint level observed in the transgenic mice, which overexpress CCN3, of two- and seven-months-old respectively. The two stains presented are valid for the structural study of cartilage, but in order to make a better comparison between the changes observed in two- and seven-months-old mice, the authors should, in my opinion, present the same staining at both ages, the one they consider most convenient, because with both of them, the structural changes described that show knee joint degeneration are observed. It would also be interesting if the images in Figures 5C, 5D and 5E were larger, at least like those in Figure 5B, for better viewing.
  • In Figure 6, in the results sections, the size of the immunohistochemical images should be larger for better viewing.
  • In the discussion section, in the paragraph between lines 277 and 280, the authors mention that the molecular mechanism by which CCN3 regulates the cell cycle is still unclear, but the authors, being prudent, should to speculate in the light of the results obtained in this work, which could be the most probable mechanism. This point should be treated more in depth in the discussion of the paper.
  • At the end of the discussion section, the authors should add a short paragraph with the most relevant conclusions based not only on those described in the literature but on their own results.
  • In the section 4.5 Immunohistochemistry, of M&M the author not mention if they done negative control in all immunohistochemical experiments. I think that yes, because in the figure 6C, the last image 74y -1st Ab, I believe that is a negative control without specific antiserum, but they should be included the specificity controls that they do in this section. The authors should be mention in this section and in the section 4.8 corresponding to western blot experiments, at which dilutions were the antisera and antibodies used.
  • In the section 4.6 of M&M the authors mention that the CCN3 antiserum was obtained immunizing rabbits with mrCCN3. Although I have already mentioned the negative control that should be included in each immunohistochemical experiment, if the authors have generated the CCN3 antiserum as it seems to be understood, they must present the corresponding specificity tests that ensure that the signal obtained in the immunohistochemical experiments is specific and corresponds to the CCN3 protein and not to another. In addition to a negative control without specific antiserum, they should face the antiserum at increasing concentrations of the CCN3 protein and use these dilutions as negative controls to verify that the intensity of the immunochemical signal decreases gradually.

Reviewer 2 Report

This topic is very interesting, and the manuscript is well written. However, after careful consideration, some concerns arise and must be addressed.

  1. Results 2.1: In Line 86, it is written that chondrocytes were isolated from E18.5 and 37-month-old mice. Is it months or weeks? The results in Figure 1 (X axis) show time in weeks. Please correct. Why the authors selected this age to represent the “aged” population. According to Jackson Laboratory, a 37-week mouse is not even considered as “middle-aged” (10-14 months) with absence of senescent changes. Therefore, these results cannot be presented for “aged chondrocytes”. Please comment.

Since the authors mentioned that Gapdh expression is decreased with age, why did they use this gene for normalisation and not Tmem199? Please justify.

It was found that Acan expression is decreased with age and Col2a1 is stable in normal mice. This contradicts with other studies, e.g. PMID: 21972019, PMID: 11795983. Please comment.

How the authors explain the opposite result of Sox9 expression using Gapdh and RNA amount? Please comment.

IHC pictures in Figure 1 are too blurred and inconclusive. Please improve quality and resolution. Negative controls should also be included.

Were the same samples used for qPCR analyses in Figure 1? The number of samples (dots) seem inconsistent between A, B etc as well as the time-points between A and the rest images in the panel.

Please clarify the number of animals used for each time-point.

  1. Results 2.2: The authors wrote that “According to current knowledge, normal articular chondrocytes in elderly knee joints rarely divide in vivo [20]. To mimic this condition in vitro, rat chondrosarcoma-derived cells (RCS) were exposed to H2O2…”. How does the addition of H2O2 mimics this condition (division)? Please comment. Furthermore, senescence should be assessed by senescence-associated β-galactosidase staining. It is known that small fraction of cells can recover from a single H2O2 stress and re-enter cell cycle, they will multiply and eventually overtake the induced premature senescent cells until contact inhibition occurs. This can be largely avoided by treating the cells with a second H2O2

Also, the authors should interpretate the result that Acan and Mmp13 expression is significantly reduced at the high H2O2 concentration added. What is the purpose of the Western blot for CCN3 at 48h? Please justify.

Finally, the senescent phenotype is characterised by the upregulation of cell cycle regulators, including p16INK4A, p21, and p53 but besides gene expression analysis, Western blots should be performed to verify protein levels.

  1. Results 2.3 and 2.4: Why RCS cells were used and not primary chondrocytes since they were isolated? At which concentration rCCN3 was added to RCS cells? How this concentration was selected? Is it a dose-dependent event?
  2. Results 2.5: IHC pictures in Figure 5 are too blurred and inconclusive. Please improve quality and resolution. Negative controls and histological scoring should also be included. What happens concerning OA progression in older mice (>7 months)?

  1. Results 2.6: Is Gapdh expression increased by age in humans like in mice? Please comment. How many human tissues were used in each age group? Please indicate.

  1. Discussion, Lines 260-261: The most prominent cytokine of the SASP is IL-6. It is unclear whether IL-6 was used as senescence marker or as a pro-inflammatory OA-associated indicator. Please comment.

  1. Materials and Methods: It is too vague whether the results in Figure 1 concern articular or ribcage chondrocytes. For example, in Figure 1A, the Ccn3/Gapdh is shown for E18.5 and this is supposed to be for articular chondrocytes while in Line 289 it is written that articular chondrocytes were isolated from postnatal day 2. Please clarify and most importantly indicate the N numbers in each experiment. Overall, the discussion is poorly written and does not contain a clear interpretation of the results. Furthermore, other studies have shown that CCN3 has a protective role in OA (PMID: 29332174, PMID: 31454155, PMID: 25541297) and should be discussed.

Round 2

Reviewer 1 Report

The authors have taken into account and fully incorporated all the suggestions made in order to improve the presentation and discussion of the results obtained.
The only change that I suggest is that section 5 of the conclusions that they have introduced, put it as section 4 after the discussion because otherwise it goes unnoticed.
In my opinion, the paper is now suitable for publication.